# Skill Preferences: Learning to Extract and Execute Robotic Skills from Human Feedback

**Xiaofei Wang**, **Kimin Lee**, **Kourosh Hakhamaneshi**, **Pieter Abbeel**, and **Michael Laskin**

University of California, Berkeley

**Abstract:** A promising approach to solving challenging long-horizon tasks has been to extract behavior priors (skills) by fitting generative models to large offline datasets of demonstrations. However, such generative models inherit the biases of the underlying data and result in poor and unusable skills when trained on imperfect demonstration data. To better align skill extraction with human intent we present Skill Preferences (SkiP), an algorithm that learns a model over human preferences and uses it to extract human-aligned skills from offline data. After extracting human-preferred skills, SkiP also utilizes human feedback to solve downstream tasks with RL. We show that SkiP enables a simulated kitchen robot to solve complex multi-step manipulation tasks and substantially outperforms prior leading RL algorithms with human preferences as well as leading skill extraction algorithms without human preferences.

**Keywords:** Reinforcement Learning, Skill Extraction, Human Preferences

## 1 Introduction

Deep reinforcement learning (RL) is a framework for solving temporally extended tasks that has resulted in a number of breakthroughs in autonomous control including mastery of the game of Go [1, 2], learning to play video games [3, 4, 5], and learning basic robotic control [6, 7]. However, today's RL systems require substantial manual human effort to engineer rewards for each task which comes with two fundamental drawbacks. The human effort required to design rewards is impractical to scale across numerous and diverse task categories and the engineered rewards can often be exploited by the RL agent to produce unintended and potentially unsafe control policies [8, 9, 10]. Moreover, it becomes increasingly difficult to design reward functions for the kinds of complex tasks with compositional structure often encountered real-world settings. In this work, we are interested in the following research question - *how can we learn robotic control policies that are aligned with human intent and capable of solving complex real-world tasks?*

Human-in-the-loop RL [11, 12, 13] has emerged as a promising approach to better align RL with human intent that proposes an alternate approach to traditional RL algorithm design. Rather than manually engineering a reward function and then training the RL agent, human-in-the-loop RL proposes for humans to provide feedback interactively to the agent as it is training. This paradigm shift sidesteps reward exploitation by providing the RL algorithm immediate feedback to align it best with human intent and, if efficient in terms of human labels required, has the potential to scale RL training across a diverse variety of tasks more reliably than reward engineering.

So far human-in-the-loop RL systems have been used to play Atari games [12], solve simulated locomotion and manipulation tasks [11, 13], and better align the output of language models [14]. While these initial results have been promising, human-in-the-loop methods are still out of reach for the kinds of long-horizon compositional tasks that are desired for real-world robotics. The primary reason is that current methods do not scale efficiently with respect to human labels for more chal-

---

Correspondence: w.xf@berkeley.edu. Project page: `https://sites.google.com/view/skill-pref`.

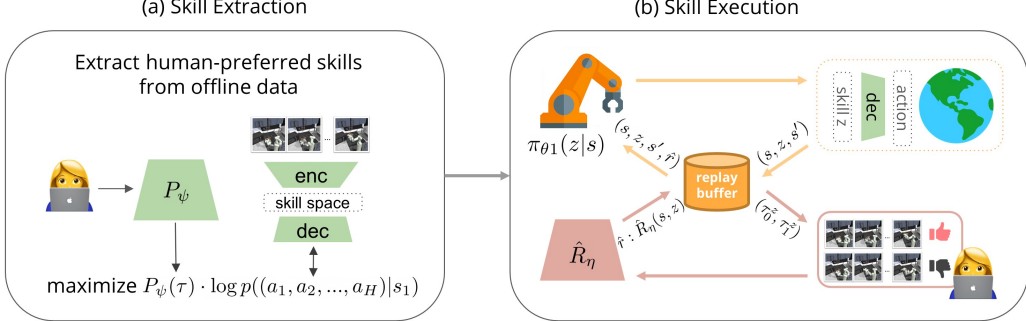

Figure 1: Our method - Skill Preferences (SkiP) - consists of two phases. During the skill extractions phase, human feedback is used to learn skills. During the skill execution phase, human feedback is used to finetune the skills to solve various downstream tasks. First, skills are extracted from a noisy offline dataset with human feedback. Second, skills are executed with RL in the environment with task-specific human feedback.

lenging tasks. As task complexity increases, the number of human feedback interactions required to attain a suitable policy becomes impractical.

To address the ability of RL algorithms to scale to more complex long-horizon tasks, a number of recent works [15, 16, 17] have proposed data-driven extraction of behavioral priors, which we refer to as *skills*. In these methods, a behavioral prior is fit to an offline dataset of demonstrations and is then used to guide the RL policy to solve downstream tasks by regularizing it to stay near the behavioral distribution. Such methods have been shown to successfully solve tasks such as diverse object manipulation [16] and operating a kitchen with a robotic arm [15]. However, they still require engineered rewards for the downstream tasks and, more importantly, assume access to a clean offline dataset of expert demonstrations that are specifically relevant to the downstream tasks. In real-world scenarios, such clean datasets are highly unlikely to exist. We desire skill extraction methods that are robust to noisy datasets, collected by a range of policies, with highly multi-modal structure.

In this work, we introduce Skill Preferences (SkiP), an algorithm that integrates human-in-the-loop RL with data-driven skill extraction. Our main insight is that human feedback can be incorporated not only for downstream RL, as is done in prior work, but also for extracting human-aligned skills. SkiP learns a human preference function and uses it to weigh the likelihood of trajectories in the offline dataset based on their degree of alignment with human intent. By incorporating human feedback during skill extraction, SkiP is able to extract structured human-preferred skills from noisy offline data and addresses the core limitation of prior skill extraction approaches - the dependence on curated expert datasets. SkiP is both capable of efficiently extracting skills and solving different downstream tasks with respect to human labels. Similar to how prior work in human-in-the-loop RL suggested replacing manually engineered reward functions with human feedback, our work suggests to replace the manual effort needed to curate clean offline datasets with human feedback. We summarize our main contributions below:

1. We introduce Skill Preferences (SkiP), an algorithm that incorporates human feedback to extract skills from offline data and utilize those skills to solve downstream tasks.

2. We show that, unlike prior leading methods for data-driven skill extraction, SkiP is able to extract structured skills from noisy offline datasets.

3. We show that SkiP is able to solve complex multi-step manipulation tasks in robotic kitchen environment substantially more efficiently than prior leading human-in-the-loop and skill extraction baselines.

## 2   Background

**Reinforcement Learning:** As is common with RL methods, we assume that the control process is a Markov Decision Process (MDP) with discounted returns. Such MDPs are defined by the tuple $\mathcal{M} = (\mathcal{S}, \mathcal{A}, R, \rho_0, \gamma)$ consisting of states $s \in \mathcal{S}$, actions $a \in \mathcal{A}$, rewards $R = R(s, a)$, an initial

state distribution $s_0 \sim \rho_0(\cdot)$, and a discount factor $\gamma \in [0, 1)$. A control policy maps states to actions within the MDP and usually takes the form of a probability distribution $-a \sim \pi(\cdot|s)$. The value function $V^\pi(s)$ and action-value function $Q^\pi(s, a)$ describe the value with respect to future expected returns with respect to an initial state or state-action pair.

$$V^\pi(s) := \mathbb{E}_{\mathcal{M}, \pi}\left[\sum_{t=0}^{\infty} \gamma^t R(s_t, a_t) \mid s_0 = s\right], \quad Q^\pi(s, a) := R(s, a) + \gamma \mathbb{E}_{s' \sim T(\cdot|s, a)}\left[V^\pi(s')\right],$$

where the first expectation $\mathbb{E}_{\mathcal{M}, \pi}$ denotes actions are sampled according to $\pi$ and future states are sampled according to the MDP dynamics. The goal in RL is the learn the optimal policy:

$$\pi^* \in \arg\max_\pi \; J(\pi, \mathcal{M}) := \mathbb{E}_{s \sim \rho_0}\left[V^\pi(s)\right].$$

In addition to the standard MDP setting, our method will also learn skills $z \in \mathcal{Z}$ which consist of an encoder that maps state-action sequences to a skill $q^{(e)}(z|s_t, a_t, \ldots, s_{t+H-1}, a_{t+H-1})$ and a decoder that maps state-skill pairs to atomic actions $q^{(d)}(a_1, a_2, ..., a_H|s, z)$.

# 3   Method

The two primary contributions of SkiP are (i) introducing human feedback during the skill extraction process to learn structured skills from noisy data and (ii) utilizing human preferences over skills for downstream RL training. Our approach shown schematically in Fig. 1 and detailed in full in Algo. 1. Due to utilizing human feedback to learn the behavioral prior, unlike prior approaches of skill extraction from offline data [17, 16, 15], our method is robust to suboptimal or noisy data.

**The SkiP Algorithm:** We first summarize the algorithm and then proceed with its derivation. Shown in Algo. 1, SkiP consists of two phases - (i) *skill extraction* and (ii) *skill execution*. A human teacher provides feedback during *both* phases. During skill extraction, a human teacher labels whether a trajectory is preferred or not (for details see Sec. 4) to train a preference classifier. A behavioral prior is then fit to the offline data with a weighted human preference function. During skill execution, the learned skills are rolled out by an RL agent - a Soft Actor-Critic (SAC) [18] - that is trained with task-specific human preferences. As such, human feedback is used during both phases of the algorithm. We proceed to define notation and provide a derivation.

**Preliminaries and Notation:** Our method is composed of two phases - (i) the skill extraction phase and (ii) the skill execution phase. During the skill extraction phase, we are given an offline dataset $\mathcal{D}$ which consists of task-agnostic, multi-modal, and potentially noisy demonstrations. We denote trajectory sequences as $\tau_t = (s_t, a_t, \ldots, s_{t+H-1}, a_{t+H-1})$, action sequences as $\mathbf{a}_t = (a_t, \ldots, a_{t+H-1})$, and skills which decode into action sequences as $z \in \mathcal{Z}$.

**Learning Behavioral Priors with Human Feedback (Skill Extraction):** Our main insight is to use human preferences in order to fit a weighted behavioral prior over an offline dataset of (potentially noisy) demonstrations. Our method builds on prior work for behavioral extraction from offline data via expected maximum likelihood latent variable models [17, 16, 15].

Specifically, prior work [17, 16, 15] considers a parameterized generative model $p_\alpha(\mathbf{a}_t|s_t)$ over action sequences where $\mathbf{a}_t = (a_t, \ldots, a_{t+H-1})$ that represents a behavioral prior and is trained to replicate the transition statistics in the offline dataset:

$$p_\alpha \in \arg\max_\alpha \mathbb{E}_{\tau \sim \mathcal{D}}\left[\sum_{t=0} \log\left(p_\alpha(\mathbf{a}_t|s_t)\right)\right]. \tag{1}$$

In our approach, we consider an adaptive behavioral prior that is biased towards trajectories that achieve higher rewards according to the human preference function. This can be particularly useful in diverse datasets collected with suboptimal or noisy policies or multiple policies of varying expertise. For example, one could imagine multiple humans collecting demonstrations or multiple robots exploring their environment. Similar to Siegel et al. [19], we seek a behavioral prior that is biased towards the high reward trajectories in the dataset while also staying close to the average statistics in the dataset. However, unlike prior work on weighted behavioral priors [19, 20, 21] the weight is determined through the human preference function and we aim to maximize action-sequence likelihood as opposed to single-timestep actions.

**Algorithm 1** SkiP: Skill Preferences

==== **Skill Extraction Phase** ====
INPUT: offline dataset $\tilde{\mathcal{B}}$
Initialize prior $p$, skill encoder $q_{\phi_2}$ and skill decoder $p_{\phi_1}$. Initialize learned preference classifier $P_\psi$
A human provides labels $(y_1, y_2, ...)$ for 10% of the trajectories in $\tilde{\mathcal{B}}$ and stores them in a new buffer $\tilde{\mathcal{D}}$
**for** each iteration **do**
    Update $\psi$ by maximizing $\mathbb{E}_{(y,\tau)\sim\tilde{\mathcal{D}}}[y \cdot \log P_\psi(\tau) + (1-y) \cdot \log(1 - P_\psi(\tau))]$
**for** each iteration **do**
    Update $p, q_{\phi_2}, p_{\phi_1}$ by optimizing $\mathcal{L}^{prior}$ (3)          {Update preference weighted behavioral prior}
==== **Skill Execution Phase** ====
Initialize parameters of actor $\pi_{\theta 1}$, critics $Q_{\theta 2}$ and $Q_{\bar{\theta} 2}$ and reward model $\widehat{R}_\eta$
Initialize a dataset of preference $\mathcal{D} \leftarrow \emptyset$ and a dataset of transitions $\mathcal{B} \leftarrow \emptyset$
**for** Each iteration **do**
    **for** Each environment step **do**
        $z_t \sim \pi(z_t|s_t), s_{t+H} \sim p(s_{t+H}|s_t, z_t), \mathcal{B} \leftarrow \mathcal{B} \cup (s_t, z_t, \widehat{R}_\eta(s_t, z_t), s_{t+H})$
    **if** iteration % $K$ == 0 **then**
        **for** step t = 1...M **do**
            $(\tau_0^{(z)}, \tau_1^{(z)}) \sim \mathcal{B}$, query human for label $y$, $\mathcal{D} \leftarrow \mathcal{D} \cup (\tau_0^{(z)}, \tau_1^{(z)}, y)$          {Get preference labels}
        **for** each gradient step of $\widehat{R}_\eta$ **do**
            Sample $(\tau_0^{(z)}, \tau_1^{(z)}, y) \sim \mathcal{D}$, update $\widehat{R}_\eta$ with min $\mathcal{L}^{reward}$ (7)          {Update preferences}
        Relabel entire replay buffer $\mathcal{B}$ with $\widehat{R}_\eta$
    **for** each gradient step of agent **do**
        Sample $(s, a, s', R) \sim \mathcal{B}$, update $\pi_{\theta 1}$ by optimizing $\mathcal{L}^{SAC}_{actor}$ (Appendix B.1)          {Update agent}
        Update $Q_{\theta 2}$ and $Q_{\bar{\theta} 2}$ by optimizing $\mathcal{L}^{SAC}_{critic}$ (Appendix B.1)

We formulate this as:

$$p_\alpha \in \arg\max_\alpha \mathbb{E}_{\tau\sim\mathcal{D}} \left[ \sum_{t=0}^{|\tau|} \omega(\tau_t) \cdot p_\alpha(\mathbf{a}_t|s_t) \right] \text{ such that } \mathbb{E}_{\tau\sim\mathcal{D}}\left[D_{KL}\left(p_\alpha \| \bar{p}\right)\right] \leq \delta, \quad (2)$$

where $\bar{p}$ denotes the empirical behavioral policy and $\omega(s_t, a_t)$ is the weighting function. The non-parametric solution to the above optimization is given by:

$$p_\alpha(\mathbf{a}_t|s_t) \propto \bar{p}(\mathbf{a}_t|s_t) \cdot \exp\left(\omega(\tau_t)/\eta\right),$$

where we have used $\propto$ to avoid specification of the normalization factor, and $\eta$ represents a temperature parameter that is related to the constraint level $\delta$. The above non-parametric policy can be projected into the space of parametric neural network policies as [20, 19]:

$$p_\alpha \in \arg\max_\alpha \mathbb{E}_{\tau\sim\mathcal{D}} \left[ \sum_{t=0}^{|\tau|} \exp\left(\omega(\tau_t)/T\right) \cdot \log\left(p_\alpha(\mathbf{a}_t|s_t)\right) \right]. \quad (3)$$

For the choice of the weighting function, we use the learned preference classifier $P_\psi(y|\tau)$ which inputs a trajectory and outputs the likelihood of this trajectory being human-preferred with $y \in [0, 1]$. $P_\psi(y|\tau)$ is learned by sampling a small subset of the offline dataset and soliciting human feedback to label preferred versus not preferred trajectory: $\omega(\tau_t) := \log P_\psi(\tau_t)$.

In this process, we treat the temperature $\eta$ as the hyper-parameter choice. This implicitly defines the constraint threshold $\delta$, and makes the problem specification and optimization more straightforward. For our practical implementation, we fit a variational autoencoder similar to [17, 15] but softly weighted to maximize the likelihood of human-preferred transitions. We introduce a latent variable $z$ with a Gaussian prior such that the ELBO loss is given by:

$$\log p(\mathbf{a}_t|s_t) \geq \mathbb{E}_{\tau\sim\mathcal{D}, z\sim q_{\phi_2}(z|\tau)} [\underbrace{\log p_{\phi_1}(\mathbf{a}_t|s_t, z)}_{\mathcal{L}_{rec}} + \beta \underbrace{(\log p(z) - \log q_{\phi_2}(z|\tau))}_{\mathcal{L}_{reg}}]. \quad (4)$$

This is the standard $\beta$-VAE loss applied to action sequence modeling where $\beta$ is a scalar controlling the regularization strength and $\phi_1, \phi_2$ are neural network parameters that are optimized during training. Note that $q_{\phi_2}$ encodes trajectories into a latent vector and $p_{\phi_1}$ decodes latent vectors and the

starting state back into action sequences. Our training objective weighs this loss with the preference function. Thus, our overall skill extraction objective is to maximize:

$$\mathcal{L} = \arg\max_{\phi_1, \phi_2} E_{\tau \sim \mathcal{D}, z \sim q_\phi(z|\tau)} \left[ P_\psi(\tau)(\mathcal{L}_{\text{rec}} + \mathcal{L}_{\text{reg}}) \right]. \tag{5}$$

**Reward learning and human preferences over skills (Skill Execution):** Unlike traditional RL where the hand-engineered rewards are available, we consider the preference-based RL framework [11, 12, 13, 22]: a (human) teacher provides preferences between the agent's behaviors and the agent uses this feedback to perform the task. In order to incorporate human preferences into deep RL, Christiano et al. [11] proposed a framework that learns a reward function $\widehat{R}_\eta$ from preferences. In this work, we modify the preference framework to operate not over atomic state-action transitions but rather state-skill transitions that have substantially longer time spans.

Formally, we assume access to an offline dataset (the agent's replay buffer) $\mathcal{B}$ of state-action transitions and sample state-skill sequence pairs $\tau_1^{(z)}, \tau_2^{(z)}$ for which a human provides a binary label $y \in \{0, 1\}$, where $\tau^{(z)} = (s_t, z_t, s_{t+H}, z_{t+H}, \ldots, s_{(t+M)H}, z_{(t+M)H})$ where $H$ is the length of actions the skill decodes to and $M$ is the total number of state-skill transitions. Note how such trajectories are $H$ times longer than if we were to sample state-action trajectories of length $M$.

The reward function $\widehat{R}$ therefore fits a Bernoulli distribution across sequences. In this work, we learn a parameterized reward function $\widehat{R}_\eta$ as in [13] utilizing a Bradley-Terry model [23] in the following manner:

$$P_\eta[\tau_1^{(z)} \succ \tau_0^{(z)}] = \frac{\exp \sum_t \widehat{R}_\eta(s_t^1, z_t^1)}{\sum_{i \in \{0,1\}} \exp \sum_t \widehat{R}_\eta(s_t^i, z_t^i)}. \tag{6}$$

Here, the operator $A \succ B$ means that $A$ is preferred to $B$. $\widehat{R}_\eta$ can therefore be interpreted as a binary preference classifier where labels are provided through human feedback. The parameters $\eta$ of the neural network are updated by optimizing a binary cross-entropy loss:

$$\mathcal{L}^{\texttt{Reward}} = -\mathbb{E}_{(\tau^0, \tau^1, y) \sim \mathcal{D}} \left[ y(0) \log P_\eta[\tau_0^{(z)} \succ \tau_1^{(z)}] + y(1) \log P_\eta[\tau_1^{(z)} \succ \tau_0^{(z)}] \right]. \tag{7}$$

## 4 Experimental Setup

**Environments:** For our experiments, we use the robot kitchen environment and offline dataset from the D4RL suite [24]. This environment requires a 7-DOF (6-DOF arm and 1-DOF gripper) robotic arm to solve complex multi-step tasks in a kitchen. Due to the 7-DOF control and compositional long-horizon nature of the tasks, this environment cannot be solved by standard methods such as SAC or behavior cloning [15].

**Offline dataset:** We desire our method to work on suboptimal offline data and, unlike prior skill extraction approaches [15, 16, 17] do not assume that the offline dataset consists solely of expert demonstrations. We simulate a noisy offline dataset by combining 601 expert trajectories and 601 noisy trajectories generated by random policy. The expert trajectories involve various structured kitchen interactions such as opening the microwave and operating the stove. We solicit human feedback on 10% of the total trajectories or equivalently 120 human labels .

**Downstream tasks:** We use 6 different downstream tasks shown in Fig. 2 that vary in difficulty to evaluate our approach. The task suite consists of tasks that require one, two, or three subtasks to be completed in a row in order to achieve the overall goal. We note that even the tasks with one subtask is challenging for RL methods that operate over atomic actions and do not leverage skills, as is shown in the experimental results.

**Simulated human:** Similar to prior work [11, 13], we obtain feedback from simulated human teachers instead of real humans. During skill extraction, human provides labels whether a trajectory is noisy or structured.[1] During skill execution, the simulated human assigns positive labels to trajectory segments that have made more progress toward completing the desired task. Progress is calculated by computing $||s_{M \cdot H} - \bar{s}||_2 - ||s_1 - \bar{s}||_2$, where $\bar{s}$ is the state when the target task is completed.

---

[1]Here, we remark that limited number of human labels (10% of the total trajectories) is utilized in our experiments for skill extraction.

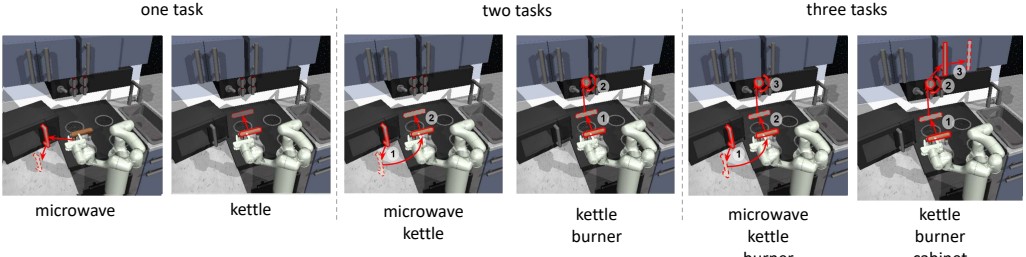

Figure 2: We evaluate in the robot kitchen environment from D4RL [24], which requires a 7-DOF robotic arm to operate a kitchen. Within this environment, we consider a variety of manipulation tasks of varying difficulty. The simplest tasks involve one subtask - opening a microwave or moving the kettle - while more challenging tasks require the agent to compose multiple subtasks. Overall, we consider 6 evaluation tasks that require chaining one, two, or three subtasks.

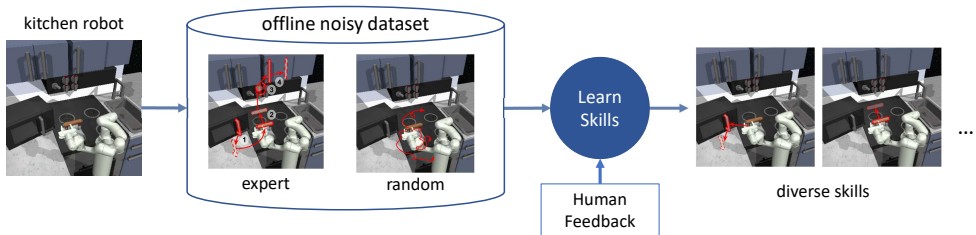

Figure 3: An illustration of the skill extraction procedure within the robot kitchen environment. Starting with a noisy offline dataset, which consists of both expert and random actions, our method fits a behavioral prior to the offline data using human feedback to identify human-preferred motions which results in a set of diverse skills that can then be finetuned to downstream tasks.

**Baselines:** In addition to our method, we compare to *Atomic Preferences* which we based on PEB-BLE [13]: a state-of-the-art human preference RL method. it pretrains the SAC agent with behavior cloning over the optimal offline dataset and trains the online SAC agent with human preferences over atomic transitions instead of high-level skill transitions. We also compare to *Flat Prior* which learns a single-step action prior on the atomic action space over the optimal dataset and trains an online SAC agent regularized with the action prior over ground-truth reward. The Oracle we compare to is *SPiRL*, a leading skill extraction with access to the ground truth (expert demonstrations and ground truth reward) in Fig. 4.

## 5 Experimental Results

For the experimental evaluation of our approach, we investigate the following questions: (a) Can SkiP solve challenging long-horizon tasks and how does our method compare to prior leading approaches? (b) How do SkiP compare to an oracle baseline that extracts skills from perfect expert demonstrations and has access to the ground truth reward? (c) Is it necessary to provide human feedback during skill extraction or is it sufficient to fit an unweighted behavioral prior over the offline data? (d) How should we incorporate human feedback during the skill execution phase?

**Main Results:** We evaluate SkiP and related baselines on the 6 tasks shown in Fig. 2 and display the learning curves in Fig. 4. We observe that SkiP is the only method (except for the Oracle) that is capable of solving the majority of tasks in the robot kitchen task suite and outperforms the baselines on all environments. On **5** out of **6** tasks, SkiP is able to match the oracle baseline asymptotically which means that it arrives at the optimal solution.

SkiP is also human-label efficient. During skill extraction, only 120 labels are required to train the preference classifier. During skill execution, 300-1K labels are required to solve most tasks depending on the task's complexity. We hypothesize that human label efficiency is better during the skill extraction phase because classifying structured and noisy skills from a static offline dataset is

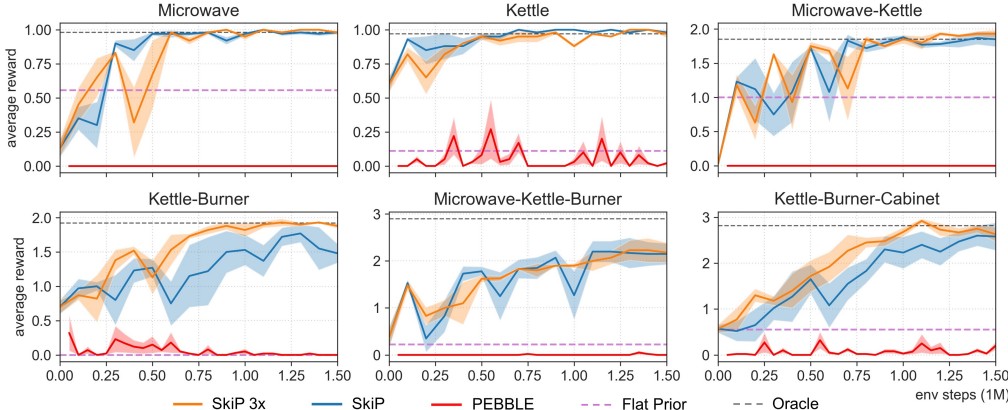

Figure 4: SkiP and baselines (Sec. 4) evaluated over six tasks in the robot kitchen environment shown in Fig. 2. SkiP outperforms both baselines across the majority of the tasks and is the only method that is capable of matching the Oracle on most tasks. We also compare SkiP to SkiP with 3x more human labels and find comparable performance between the two versions. SkiP solves most tasks given 300-1000 human labels depending on the complexity of the task.

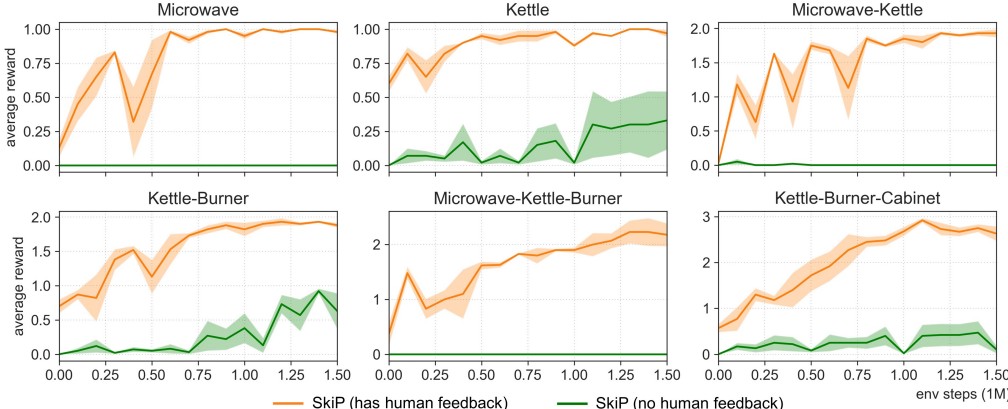

Figure 5: SkiP with human feedback vs SkiP without human feedback during skill extraction. learning curve with shaded region representing standard error across three seeds. Both algorithms learns prior from the suboptimal dataset and were evaluated with online RL. SkiP with human feedback outperforms SkiP without human feedback on all 6 environments

easier than classifying task-specific preferences from an evolving replay buffer. Further human label efficiency improvements pose interesting research directions for future work.

**Ablation Studies:** To further understand the properties of the SkiP algorithm, we investigate whether human feedback is necessary during skill extraction as well as how the human preference reward function compares to alternate approaches to human feedback during skill execution.

*Is it necessary to provide human feedback during skill extraction or is it sufficient to fit an unweighted behavioral prior over the offline data?* The offline dataset used throughout this paper consists of suboptimal data that is a mixture of expert and random actions. We compare fitting a human-feedback weighted behavioral prior as opposed to an unweighted behavioral prior that maximizes the likelihood of all action sequences equally. For the skill execution phase, both methods have access to the same human preference reward function. The results shown in Fig. 5 indicate that the method, which extracts skills without human feedback, is unable to solve any of the tasks suggesting that human feedback is essential for skill extraction from suboptimal offline data.

*How should we incorporate human feedback during the skill execution phase?* Instead of preferences, a simpler approach to learning from human feedback is to provide binary feedback if a task (or subtask) has been solved and learning a reward classifier to guide the RL agent. We implement

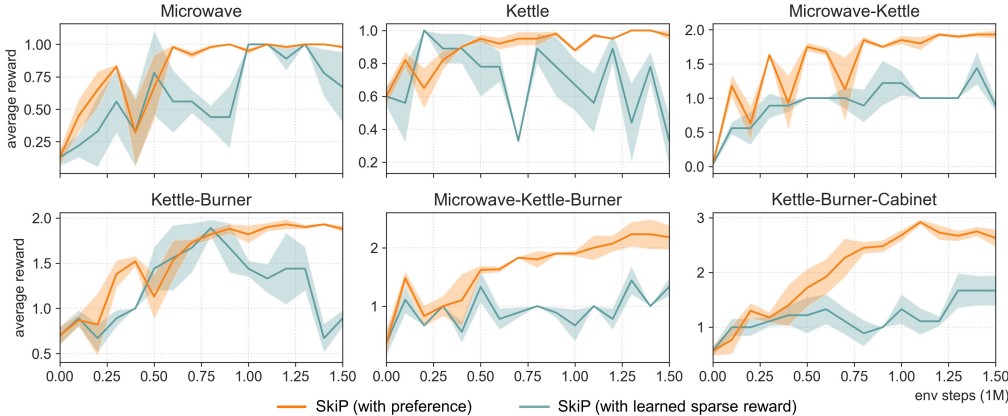

Figure 6: SkiP with preferences vs SkiP with learned sparse reward. Learning curve with shaded region representing standard error across three seeds. Both algorithms use the same prior. SkiP with preferences outperforms SkiP with learned sparse reward on 5 out of 6 environments.

this by providing a positive reward of 1 for a high-level transition $(s_t, z, s_{t+H})$ when a subtask has been completed and 0 otherwise. Using the same number of human queries for both approaches, we compare learning with preferences as opposed to learning from sparse rewards. For both approaches, we use human feedback for skill extraction. As shown in Fig. 6, RL with a reward classifier for subtask completion is able to solve some tasks but generally performs much worse than RL with human preferences.

## 6   Related Work

**Human-in-the-loop Reinforcement Learning:** Several works have successfully utilized feedback from real humans to train RL agents [25, 11, 12, 26, 13, 27, 28]. One of major directions is directly utilizing the human feedback as a learning signal [29, 27, 25] but assumed unlimited access to human labels which limited their practicality for more challenging tasks. To address this limitation, a number of works proposed learning reward model from human feedback [26, 28, 30, 31, 32, 33]. Recently, several works have successfully combined human preferences with deep RL algorithms to learn basic locomotion skills as well as playing video games from pixels using human [11, 12, 34, 13]. However, these methods are limited to short-horizon or cyclic tasks and do not scale to more challenging compositional multi-step tasks. In this work, we investigate how to scale human preferences to such challenging tasks by specifying preferences over skills.

**Data-driven Extraction of Behavioral Priors:** Behavioral prior or skill extraction refers to fitting a distribution over an offline dataset of demonstrations and biasing the agent's policy towards the most likely actions from that distribution. Commonly used for offline RL [21, 19, 20], behavioral priors learned through maximum likelihood latent variable models can also been used as skills for structured exploration in RL [16], to solve complex long-horizon tasks from sparse rewards [15, 17], and regularize offline RL policies [21, 20, 35]. A limitation of these skill extraction methods is that the quality of the behavioral prior is highly dependent on the demonstrations in the offline dataset. Since a behavioral prior models maximum likelihood transitions in the offline dataset, suboptimal, noisy, or irrelevant transitions can degrade downstream policy learning. In this work, we introduce human feedback into the skill extractions phase to learn a human preferred behavioral prior which enables skill extraction methods to be robust to suboptimal offline data.

## 7   Conclusion

We presented Skill Preferences (SkiP) an algorithm that uses human feedback for both skill extraction as well as execution, and showed that SkiP enables robotic agents to solve long-horizon compositional manipulation tasks. We hope that this work excites other researchers about the potential of learning with skills and human feedback.

## 8 Acknowledgements

We would like to thank Berkeley DeepDrive, Tencent, ONR Pecase N000141612723, and NSF NRI 2024675 for supporting this research.

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
