# OpenReview forum: "Skill Preferences: Learning to Extract and Execute Robotic Skills from Human Feedback"
_robot-learning.org/CoRL/2021/Conference — CoRL2021 Poster_

### Official Review · Reviewer_6ucT · 2021-07-02

**Originality:** Very Good
**Technical Quality:** Good
**Clarity Of Presentation:** Good
**Impact:** 4

**Recommendation:**

Weak Accept: I recommend accepting the paper, but will not argue for my recommendation if the majority of other reviewers have a different opinion.

**Summary:**

The paper proposes a deep RL algorithm that jointly leverages human preference feedback and offline datasets of agent experience (via the extraction of learned skills) to enable the learning of long-horizon tasks. As a result, it can learn longer and more complex tasks than prior human-in-the-loop RL papers, and better cope with mixed offline data quality than prior skill-based RL approaches.

**Issues:**

## Suggestions to improve the paper
- In order to address (A) I would suggest collecting a more realistic mixed-quality dataset, e.g. by training a BC policy on the expert data and using it to collect lower-quality trajectories for training (they will be lower quality due to accumulating errors) --> this technique was used in SPiRL [15] to investigate robustness to lower-quality data. Alternatively one could also add action-noise to the demonstration actions or to rollouts from a trained oracle SPiRL policy to generate lower-quality sequences that are **not** completely random. By combining these sequences with the higher-quality expert sequences the proposed approach could be evaluated in a more realistic mixed-quality setting.

- With lower priority: to address (B) one could add evaluation of the idea on another environment, e.g. any of the other environments used in the SPiRL paper, to show that the proposed approach is applicable in flexible environments.

- It would be interesting to add a baseline to the comparison in Fig 6 that uses the SkiP skills with **true** sparse subtask reward (ie does not learn to estimate the subtask reward but uses the GT environment reward) --> this could show how much performance we loose from the reward estimation (also in comparison to the results in Fig 4)

**Reviewer Expertise:**

Excellent: Expert knowledge on the topic of the paper

**Strengths And Weaknesses:**

## Strengths
- **The intuition of the method is convincing**: human feedback is used in two separate ways by the approach: (1) to guide skill extraction towards human skill preferences, (2) to provide feedback during downstream learning. Both cases are well motivated in my opinion:
    (1) Generally, we cannot assume that the offline experience dataset used for skill pre-training contains only optimal sequences. At the same time we cannot assume that it features reward annotations for the target task. Thus, we lack a signal to determine which of the sequences are high quality and should be used for learning skills. Using human feedback for this choice, as proposed by the paper, is a particularly intuitive choice since humans are able to provide an estimate of the desirability of a motion sequence irrespective of the particular downstream task.
    (2) Using preferences for downstream task learning is not a new idea, there is a long line of work trying to learn rewards from preferences (see related work section). One critical issue is that early in training, when mostly random trajectories get compared to one another, it is hard to formulate effective preferences and training is slow. Thus prior work has used unsupervised pre-training too jump-start this process (PEBBLE, Lee et al. 2021). Here, the unsupervised pre-training is replaced with pre-training from demonstration-like offline data which enables preference-based learning on substantially more complex and longer-horizon tasks -- an extension which makes sense to me.

- The approach is technically simple: given the problem setting the chosen technical approach is rather straightforward but seems to achieve good empirical performance. Thus, I would see its simplicity as a strength.

- The experimental evaluation shows the benefits of the approach, both in using human feedback for skill extraction (Fig 5) and using learned skills for preference-based RL (Fig 4)

- The performed experiment on preference-based reward vs (learned) sparse reward (Fig 6) is interesting -- it essentially shows that human preferences can provide a denser reward signal than the rather sparse environment subtask rewards. This might be a known result in the literature on RL from human preferences, but I am less familiar with this literature and thus found it interesting.


## Weaknesses
In my opinion, the main weakness is in the experimental evaluation of the proposed approach. In particular the dataset used for skill pre-training does not reflect a realistic mixed-quality dataset:

(A) The offline data used for skill pre-training consists to equal parts of (1) expert-level human-teleoperated sequences and (2) completely random-action trajectories. Thus it is trivial to distinguish between "good" and "bad" trajectories and, more importantly, none of the behaviors seen in the "bad" trajectories is actually relevant for solving the downstream task. In a more realistic evaluation scenario the data would instead consist of "better" trajectories that feature _mostly_ useful behavior and "worse" trajectories that contain lower quality interactions but _also_ have some reasonable behavior. It seems important to investigate whether the proposed simple approach of preference-based weighting during skill extraction would still work in such a more ambiguous scenario instead of the black-and-white scenario that was tested now, particularly since the preference-based skill extraction is the major novelty of the proposed approach.

(B) The approach is only evaluated in a single environment with a single pre-training dataset (although on different downstream tasks). Thus it lacks evaluation on different robotic agents with different observation and action spaces and different dataset characteristics. That being said: while this does make the evaluation less comprehensive, I would see this point as less severe than the point about the used dataset above, since the kitchen environment used for evaluation is rather challenging.

Minor Comments:

(C) The current submission seems a bit unpolished which sometimes impacts its clarity. While this can surely be addressed with a bit of minor rewriting, it seems important to clarify some details (see questions below) and proof-read the submission again to fix typos (eg caption to Fig 2, L134, caption to Fig 6)


## Questions
(a) Is the human feedback labeling during skill execution done per sequence or per skill segment? What are the states that get compared in the preference function in L177?

(b) Why is the "Flat Prior" in Fig 4 a flat line and not a learning curve? Since it finetunes the policy with SAC I would expect this to be a learning curve.

(c) Does the proposed approach use regularization with the skill prior during downstream RL as is done in SPiRL [15]? The update equation Algorithm 1 points to (in appendix A) is the vanilla SAC update equation that uses the non-learned uniform prior over actions (ie regularizes with a maximum entropy objective) instead of minimizing divergence to the learned skill prior. If indeed the vanilla SAC objective is used can you please elaborate on why the learned skill prior is not leveraged here?


**Summary Of Recommendation:**

Overall, I am convinced by the relevance of the problem and the intuition of the proposed approach. The technical implementation is simple and solid. I am slightly on edge due to the somewhat toyish "mixed-quality dataset" setup, but I lean towards accepting the submission, in particular if the authors can address some of the points suggested above -- focussing on a more realistic mixed-quality dataset composition.

---

### Official Review · Reviewer_XBne · 2021-07-22

**Originality:** Good
**Technical Quality:** Good
**Clarity Of Presentation:** Good
**Impact:** 3

**Recommendation:**

Weak Reject: I recommend rejecting the paper, but will not argue for my recommendation if the majority of other reviewers have a different opinion.

**Summary:**

This paper studies the problem of reinforcement learning with access to large offline (potentially diverse and noisy) demonstration datasets. Under this setting, the traditional approach is to learn behavioral priors in the form of skills (i.e. a latent var z that maps to a sequence of actions) in the first phase, and then running standard reinforcement learning algorithms on top of the learned skills in the second phase. Doing RL on top of skills, as opposed to raw actions, makes it easier to solve long-horizon tasks.

The paper operates in the above setting, and examines the scenario in which we have additional access to humans in the loop -- i.e. how can we maximally improve existing approaches if we're allowed a reasonable amount of human labels. The proposed method (SkiP), proposes to take advantage of human labels in both the first phase (skill extraction where we learn to map latent z to action sequences) and the second phase (skill execution where we run RL over skills). For skill extraction, the humans label trajectories with a likelihood score from 0 to 1, and a generative model of skills is learned with a beta-VAE objective that weighs trajectories based on the labels. For skill execution, the humans are given pairs of trajectories and choose the more preferred trajectory, which is used to recover the reward function that maximizes the probability (under the Bradley-Terry model) of the human preferences.

The paper experiments on a robot kitchen environment that involves long horizon tasks. The SkiP algorithm is tested using 100 to 1000 (simulated) human labels, showing better results than PEBBLE (which also incorporates human feedback but over individual actions instead of skills).


**Issues:**

See above.

**Reviewer Expertise:**

Good: General knowledge of the area

**Strengths And Weaknesses:**

Strengths:
- the problem addressed is important, and the proposed method is simple and clear
- learning preferences over action sequences instead of atomic actions is more efficient and natural for humans

Weaknesses:
- the initial conditions and termination conditions of the skills are not learned, instead a horizon H is picked and the total trajectory is decomposed into action sequences of length H. This seems to essentially be offloading all the work to the humans, e.g. a sequence of H actions will be rated highly if the actions make sense and also aligns with reasonable initial/terminal conditions. There seems to be a missing piece of segmenting the data into skills before presenting to the human (or even incorporating human feedback to segment the data in a smart way).
- the motivation leading up to Eq 4 and 5 is not very well presented. Eq 3 and the equation before it are presented without much intuition (looks like it's pulled from [19,20], but should probably be presented more formally e.g. as a lemma). Also, hyperparameters change from delta to T (and possibly to beta?), but the exact correspondence is not clear. It's not clear to me where T ends up affecting the final objective in equation 5. More importantly, the beta-VAE objective is also pulled without a direct connection to the derivation in Eq 3. For example, it's not obvious that putting the preference P(\tau) as a multiplier is the principled thing to do / aligns with Eq 3.

**Summary Of Recommendation:**

The proposed method is simple and tackles an interesting problem. But, as it stands, the contribution is a bit thin -- the modeling techniques are chosen without new insights and without comparison to any other choices. The paper can be improved by considering more sophisticated approaches to curating action sequences to present to the human labeler, comparing with other choices of generative models (or having a more convincing derivation of the current modeling approach e.g. why beta-VAE).

---

### Official Review · Reviewer_hYAX · 2021-07-23

**Originality:** Very Good
**Technical Quality:** Very Good
**Clarity Of Presentation:** Good
**Impact:** 4

**Recommendation:**

Strong Accept: I recommend accepting the paper and will argue for my recommendation even if other reviewers hold a different opinion.

**Summary:**

This work introduces an algorithm, SkiP, that uses preferences to learn skills from noisy demonstrations, and uses these skills along with more preference data to learn tasks using RL. This work also explores how much preference feedback is needed, as well as determining whether preferences or sparse rewards produce better learning results during skill execution. SkiP is demonstrated outperform baseline algorithms, while improving over prior methods by utilizing noisy demonstration data.

**Issues:**

Issues to address in the author response/revision period:
-	Is there any existing data that suggests how SkiP might behave if some preferences were mislabeled?
-	Is the state-skill sequence on line 147 supposed to end in s_(t+(M)H), z_(t_(M)H) instead of s_((t+M)H), z_((t+M)H), as there are M state-skill transitions of length H starting from s_t? Or perhaps I misunderstood the notation.
-	Understandably, this wasn’t tested with human teachers, but instead with simulated people. How long might it take for a human teacher to provide 120 + 300/1K preference labels to full state-skill sequence pairs?
-	Just a small note that there are a few little typos throughout the paper (vecotr instead of vector in line 134, for example)

**Reviewer Expertise:**

Good: General knowledge of the area

**Strengths And Weaknesses:**

SkiP’s main strength is its ability to extract skills from noisy demonstrations without having to have an expert define a reward function, and the results seem convincing of this claim. The experiments are well-rounded as well. This paper shows SkiP’s performance with more human data, to show that the training does not improve much with more feedback than the amount used (120 preferences/~10% of trajectories during skill extraction). SkiP is also tested to see if sparse rewards could be used rather than preferences during skill execution. It seems evident from the testing that SkiP’s formulation provides good learning over other formulation options.

SkiP does still seem to require correct preference feedback to learn well. If a person was confused about how certain skills worked, or mislabeled preferences (particularly given the 300-1K required labels during skill execution), it’s unclear how SkiP would perform. However, SkiP is still able to learn quite well from noisy demonstrations, which is very useful for skill learning.


**Summary Of Recommendation:**

I am recommending a weak accept because while SkiP is able to learn very well from noisy demonstrations, it still requires a decent amount of correct information from humans in the form of trajectory preferences. However, this work does seem like it is a good step forward, and preferences are easier to provide than demonstrations.

Update post-rebuttal: The new experiments on more realistic noisy demonstrations and the analysis of noisy preference labels have caused me to change my recommendation to strong accept (contingent on these results being added to the paper or appendix).

---

> ### Comment · Reviewer_hYAX · 2021-09-02
> **Updated Response**
>
> Thanks to the authors for their thorough response to my questions. The new experiments on more realistic noisy demonstrations, and the analysis of noisy preference labels are particularly helpful. If the experiments were added to the paper (perhaps in the appendix) I would change my recommendation to a strong accept. For now (unless I missed an update in the paper), I am leaving my recommendation at a weak accept.

---

> > ### Author Response · Authors · 2021-09-02
> > **Response to the Updated Response**
> >
> > Apologize for forgetting to put these results in the Appendix! I was too focused on revising the paper to accommodate the reviewers for this rebuttal and forgot other potential readers who would only read the paper but not the comment section here. We will be sure to put these results in the Appendix for the camera ready version if it gets accepted.

---

> > > ### Comment · Reviewer_hYAX · 2021-09-02
> > > **Updated Recommendation**
> > >
> > > Thanks for the response! With the planned addition in mind, I'm changing my recommendation to a strong accept.

---

### Official Review · Reviewer_noDs · 2021-07-24

**Originality:** Good
**Technical Quality:** Excellent
**Clarity Of Presentation:** Excellent
**Impact:** 4

**Recommendation:**

Strong Accept: I recommend accepting the paper and will argue for my recommendation even if other reviewers hold a different opinion.

**Summary:**

This paper presents a clean and elegant approach for 1) extracting skills from offline datasets of suboptimal/noisy human demonstrations and 2) using the extracted skills to solve downstream, multi-step complex robot manipulation tasks, *by intelligently leveraging a human-in-the-loop*. Critically, the proposed approach — SkiP — uses preferences from humans to learn good priors over skills that are worth extracting from offline datasets (e.g., skills that lead to higher rewards/succeed at the goal more often), and in addition, use human preferences during task learning to identify the “right” skill transitions (which skills to use) to best solve complicated tasks.

This approach is well-motivated, and the evaluation is complete, and clear. Looking at a series of D4RL tasks with a 7-DoF robot manipulator (6-DoF Arm + 1-DoF Gripper) the proposed approach SkiP not only outperforms strong approaches that 1) just extract skills without preference information and 2) are trained with human preference information, but is also label efficient from a human perspective, with clean ablations that show how SkiP performs when human-labels are removed at each of the two stages (extraction and learning).

Coupled with nice oracle experiments and visualizations, this is a nice step forward in learning and using skills from offline data with arbitrary optimality guarantees, presenting a nice step forward for long-horizon robotics.

**Issues:**

Questions
---

- [Curiosity more than anything else] Do you have a sense of how robust SkiP is to noise in the human-preference labeling process? Is there more of an impact when there’s preference noise during skill extraction (my guess) or learning?

Typos
---
Figure 1 Caption - “to denoise behavioral prior”?
131 - Guassian —> Gaussian
Figure 2 Caption - “mire challenging” —> “more challenging”

**Reviewer Expertise:**

Good: General knowledge of the area

**Strengths And Weaknesses:**

This paper is thorough and extremely clear. The strengths of this approach involve the simplicity of the two stage approach — first extract skills by using human preferences to “weight” behaviors that are more relevant/meaningful, then use human preferences again during learning to figure out the right “skill transitions” to quickly learn how to compose skills to solve complex tasks. The experiments are complete, the baselines strong, and the ablations/oracle numbers paint a clear picture of the benefits of the proposed approach.

In terms of weaknesses, I am worried about the actual performance of SkiP models in the presence of offline data that is more muddled (all demonstrations are suboptimal in some way). As I currently understand the data used to train the SkiP models, a dataset of off-line demonstrations was constructed from 50% optimal demonstrations of a task, and 50% data drawn from a random policy. I think this is a nice first step, but I don’t know how this approach would perform with noise directly injected into the optimal demonstrations (sub-optimal behaviors at times).

Separately, while I understand that this is keeping with prior work, using simulated humans for labeling is still a limitation in my opinion; I don’t think that this is specific to this work, but I would hope that future work looks to how actual humans preferences correlate with those of the simulated humans.

**Summary Of Recommendation:**

This paper proposes a straightforward solution, performs a robust and thorough evaluation, insightful analyses, and situates itself well related to prior work. While there are some weaknesses around the offline dataset used to validate the method, I believe this is strong work that could be extremely useful for the field, and propel our ability to scale manipulation to more complex, long-horizon tasks.

Would be nice to eventually get this on a robot though ;)

---

> ### Author Response · Authors · 2021-08-29
> **Response to Reviewer noDs**
>
> Thanks so much for spending a lot of time on reviewing this work and raising constructive questions!
> We are glad that you find our proposed method to be a straightforward solution and find the analysis to be insightful.
>
> **Q1:**  Performance of SkiP models in the presence of offline data such that all demonstrations are suboptimal in some way
> **A1:**  Thanks for this constructive feedback. To test our method on more realistic mixed-quality demonstration, we added an ablation that simulated imperfect demonstrations by adding Gaussian noise to some of the action sequences in the offline dataset. The scale of the Gaussian noise is 10% of maximum action magnitude. We ablate the downstream performance on the three-task Kettle-Burner-Cabinet environment against the percentage of noise present in the offline dataset. We found that the weighting during skill extraction is still mostly effective even with noisy dataset.
>
> | Percentage of noisy data     | Return at 1.5M steps| Return at 800K steps|
> | -----| ----- | -- |
> |0%  |   $2.8\pm{0.2}$   |   $2.5\pm{0.2}$ |
> |5%  |$2.7\pm{0.4}$  |  $2.3\pm{0.6}$   |
> |10%|  $2.3\pm{0.6}$  | $1.7\pm{0.7}$ |
> |20 %| $2.4\pm{0.6}$ | $1.8\pm{1.0}$|
>
>
> **Q2:** How robust SkiP is to noise in the human-preference labeling process? Is there more of an impact when there’s preference noise during skill extraction or learning?
> **A2:** Thank you for suggesting this robustness test! We ablate the downstream performance on the three-task Kettle-Burner-Cabinet environment against the percentage of random human preference labels. We find that preference noise in the skill extraction part greatly impacts downstream learning. For noise during skill execution (learning), we didn’t find there to be a notable difference between the performance without noise or with 20% of such noise. Thus, we agree that there is more of an impact for preference noise during skill extraction than during skill execution.
>
> | Percentage of random preference (extraction) | Return at 1.5M steps| Return at 800K steps|
> | ----- | ----- | -- |
> |0%  |  $2.8\pm{0.2}$   |   $2.5\pm{0.2}$ |
> |4%  | $1.9\pm{0.6}$  |  $1.6\pm{0.4}$   |
> |10%|  $1.5\pm{0.4}$  | $1.3\pm{0.8}$ |
> |20 %| $1.2\pm{0.7}$ | $1.0\pm{0.5}$|

---

> ### Comment · Reviewer_noDs · 2021-08-31
> **Brief Score Justification (Post-Rebuttal)**
>
> I found the rebuttal great! Loved that the authors cleared up my misconceptions, and even ran experiments to answer my question about preference noise! I will keep my score of Strong Accept.

---

### Meta-Review · Area_Chair_DTtm · 2021-08-12

**Recommendation:** Accept (Poster)
**Confidence:** 5

**Metareview:**

The reviewers agree that the problem addressed is important and the proposed method is simple, clear, and able to learn very well from the demonstrations. The two-phase approach is intuitive, and the experimental evaluation are convincing and show interesting results.

There are a number of concerns brought up by the reviewers that the authors should try to address during the response period, including:
1) The offline dataset is not representative of a realistic mixture of mixed-quality demonstrations, and collecting and learning from a more realistic dataset (as suggested by one of the reviewers) can demonstrate the effectiveness of the approach a bit better.
2) The definition of skills: the authors use a fixed time horizon for skills and the initial and termination conditions are not learned. It would be interesting to have an automatic way of segmenting the skills or incorporating human feedback to segment the skills. This also makes me wonder if the skills are just expert action reconstruction sequences (which makes me question the generalizability of them), and if you could simply remove the preference predictor weighting and train a standard VAE over the expert data and basically learn the same action sequences (skills).
3) There are some concerns about the contributions of the paper. Specifically the paper does not provide new insights or comparison to other modeling choices (e.g. the specific choice of generative models).
4) The approach is only evaluated in a single simulated environment, and it would be interesting to assess the effectiveness of the approach beyond this environment or on real robotic systems.

**post rebuttal**
The authors have addressed the majority of reviewers' concerns, and provided new experiments. Overall based on the reviewer's suggestions, I recommend this paper for publication.

---

### Decision · Program_Chairs · 2021-09-13

**Decision:**

Accept (Poster)

**Comment:**

The reviewers agree that the problem addressed is important and the proposed method is simple, clear, and able to learn very well from the demonstrations. The two-phase approach is intuitive, and the experimental evaluation are convincing and show interesting results.

There are a number of concerns brought up by the reviewers that the authors should try to address during the response period, including:
1) The offline dataset is not representative of a realistic mixture of mixed-quality demonstrations, and collecting and learning from a more realistic dataset (as suggested by one of the reviewers) can demonstrate the effectiveness of the approach a bit better.
2) The definition of skills: the authors use a fixed time horizon for skills and the initial and termination conditions are not learned. It would be interesting to have an automatic way of segmenting the skills or incorporating human feedback to segment the skills. This also makes me wonder if the skills are just expert action reconstruction sequences (which makes me question the generalizability of them), and if you could simply remove the preference predictor weighting and train a standard VAE over the expert data and basically learn the same action sequences (skills).
3) There are some concerns about the contributions of the paper. Specifically the paper does not provide new insights or comparison to other modeling choices (e.g. the specific choice of generative models).
4) The approach is only evaluated in a single simulated environment, and it would be interesting to assess the effectiveness of the approach beyond this environment or on real robotic systems.

**post rebuttal**
The authors have addressed the majority of reviewers' concerns, and provided new experiments. Overall based on the reviewer's suggestions, I recommend this paper for publication.